# Progresses and Pitfalls of Epigenetics in Solid Tumors Clinical Trials

**DOI:** 10.3390/ijms252111740

**Published:** 2024-10-31

**Authors:** Alice Rossi, Francesca Zacchi, Anna Reni, Michele Rota, Silvia Palmerio, Jessica Menis, Andrea Zivi, Stefano Milleri, Michele Milella

**Affiliations:** 1Section of Innovation Biomedicine-Oncology Area, Department of Engineering for Innovation Medicine (DIMI), University and Hospital Trust (AOUI) of Verona, 37134 Verona, Italy; 2Centro Ricerche Cliniche, 37134 Verona, Italy

**Keywords:** epigenetic drugs, precision medicine, anticancer treatment, solid tumors

## Abstract

Epigenetic dysregulation has long been recognized as a significant contributor to tumorigenesis and tumor maintenance, impacting all recognized cancer hallmarks. Although some epigenetic drugs have received regulatory approval for certain hematological malignancies, their efficacy in treating solid tumors has so far been largely disappointing. However, recent advancements in developing new compounds and a deeper understanding of cancer biology have led to success in specific solid tumor subtypes through precision medicine approaches. Moreover, epigenetic drugs may play a crucial role in synergizing with other anticancer treatments, enhancing the sensitivity of cancer cells to various anticancer therapies, including chemotherapy, radiation therapy, hormone therapy, targeted therapy, and immunotherapy. In this review, we critically evaluate the evolution of epigenetic drugs, tracing their development from initial use as monotherapies to their current application in combination therapies. We explore the preclinical rationale, completed clinical studies, and ongoing clinical trials. Finally, we discuss trial design strategies and drug scheduling to optimize the development of possible combination therapies.

## 1. Introduction

Epigenetics, defined as the study of changes in gene expression that do not involve alterations to the underlying DNA sequence, has become a key area of research for understanding cancer and developing new treatments. Unlike genetic mutations, which directly modify the DNA code, epigenetic changes involve chemical modifications, such as the addition of methyl groups to DNA or the modification of histone proteins around which DNA is wrapped. These changes can either activate or silence genes, influencing cellular behavior without altering the genetic code itself [1].

In the context of cancer, abnormal epigenetic modifications can lead to the inappropriate activation of oncogenes or the silencing of tumor suppressor genes, driving the uncontrolled growth characteristic of cancer cells. These epigenetic alterations are often reversible, making them attractive targets for therapeutic intervention [2]. Research aimed at reversing these abnormal modifications has led to the development of epigenetic therapies, including DNA methyltransferase (DNMT) inhibitors and histone deacetylase (HDAC) inhibitors. By reprogramming the epigenome, these treatments may restore normal gene expression patterns in cancer cells, potentially slowing disease progression and synergistically enhancing the effectiveness of existing therapies such as chemotherapy, radiation, and immunotherapy [3].

The potential of epigenetics in cancer treatment extends beyond the general approach of targeting common cancer pathways. As our understanding of epigenetics deepens, it is becoming increasingly clear that these mechanisms play a crucial role in cancer development and progression. Nevertheless, epigenetic drugs have been tested both as monotherapies and in combination treatments, with highly variable results so far. Given that the epigenome is highly dynamic and varies between individuals and even among different cells within a tumor, this variability opens the door to personalized medicine, a field in which treatments can be tailored to the specific epigenetic landscape of a patient’s tumor. Consequently, epigenetic therapies represent a promising frontier in the quest for more effective, targeted, and less toxic cancer treatments, offering hope for improved outcomes in a disease that remains one of the leading causes of death worldwide [4].

## 2. Generations of Epigenetic Drugs as Monotherapy

Genome function is regulated at multiple levels by epigenetic enzymes, and in recent years, multiple generations of epigenetic drugs have been developed (Figure 1).

First-generation epigenetic drugs include DNMT inhibitors (e.g., azacytidine and decitabine) and HDAC inhibitors (e.g., vorinostat and romidepsin). These compounds are characterized by a very broad spectrum of action against all isoforms of DNMT and HDAC and are not biomarker-driven, following the concept of the “one size fits all”. This lack of specificity represents the major issue of these drugs, as different severe toxicities have been described very frequently. This generation of drugs has shown very limited efficacy and low response rates in solid tumors, whereas, in hematological malignancies, they still represent an important option of treatment in certain settings [6].

In order to guarantee better efficacy with an acceptable safety profile, second-generation epigenetic drugs have been developed. In this family, second-generation DNMT inhibitors (e.g., guadecitabine) and second-generation HDAC inhibitors (e.g., belinostat and panobinostat) are included. These compounds are designed to interfere with only some isoforms of DNMT and HDAC and have demonstrated a much better manageability compared to first-generation inhibitors; nevertheless, efficacy has demonstrated to be largely disappointing, particularly when these agents are used as monotherapy [7]. Thus, combinations with other therapeutic strategies have been tested (chemotherapy, immunotherapy, targeted therapy, etc.), with contrasting results that will be examined in detail later. Subsequently, the concept of developing a therapeutic strategy related to precision medicine led to the development of third-generation epigenetic drugs, including, among others, histone methyltransferases (HMT) inhibitors, bromodomain and extra-terminal domain (BET) inhibitors, lysine-specific histone demethylase 1A (LSD1) inhibitors, isocitrate dehydrogenase (IDH) inhibitors, histone demethylase (HDM) inhibitors, and protein arginine methyltransferase (PRMT) inhibitors. This class of drugs resulted in improved efficacy, leading, in few cases, to the Food and Drug Administration (FDA) approval in solid tumors.

Bromodomain and extra-terminal domain (BET) proteins are structures involved in the regulations of gene expression, and their inhibition represents a promising therapeutic target. Among the BET inhibitors, Birabresib (OTX015/MK-8628) showed efficacy in patients affected by NUT midline carcinoma (NMC) harboring the BRD4–NUT fusion oncoprotein, providing the first proof-of-concept evidence of clinical activity of a BRD inhibitor in targeting BRD4–NUT; three out of ten patients (30%) with NMC had a partial response (PR) with duration of response (DOR) of 1.4 to 8.4 months [8]. The efficacy of Birabresib was then studied in patients affected by NMC with a BRD–NUT gene translocation in a phase 1b trial where Birabresib exhibited clinical activity in NMC and a favorable safety profile; however, the authors suggested that intermittent dosing schedules might help reduce the toxicities associated with chronic administration [9]. The proof of concept for the efficacy of BET inhibitors in NMC was further supported by a phase 1 study, with Molibresib (GSK525762) showing PR in 4 out of 19 patients (21%) and stable disease (SD) in 8 patients (42%) as the best response in patients affected by NMC [10].

Histone methyltransferases (HMTs) are also proteins involved in the regulation of gene expression. Tazemetostat, an inhibitor of the histone-lysine N-methyltransferase enhancer of zeste homolog 2 (EZH2), demonstrated efficacy in a phase II trial including patients with advanced epithelioid sarcoma with INI1/SMARCB1 loss: an objective response rate (ORR) was reached in 15% of patients, with a median progression-free survival (PFS) of 5.5 months (95% CI 3.4–5.9) and median OS (overall survival) of 19.0 months (11.0—not estimable) [11]. Based on these data, in 2020 the FDA granted accelerated approval to tazemetostat as the first treatment option for adults and pediatric patients with metastatic or locally advanced epithelioid sarcoma who are not candidates for complete resection [12].

Isocitrate dehydrogenase (IDH) is an enzyme involved in the Krebs cycle that catalyzes the transformation from isocitrate to alpha-ketoglutarate, an irreversible step fundamental for cell metabolism and stability; the inhibition of IDH could lead to alterations in gene expression and be a pro-oncogenic factor [13]. Among IDH mutation inhibitors, ivosidenib demonstrated efficacy in the phase III ClarIDHy trial (NCT02989857), in which 185 patients with IDH1-mutated advanced cholangiocarcinoma previously treated with platinum-based were enrolled. This study showed a statistically significant improvement in PFS with ivosidenib compared to a placebo (HR 0.37, 95% CI, 0.25–0.54, *p* < 0.001). The median OS was found out to be 10.3 months with ivosidenib versus 7.5 months with the placebo (hazard ratio: 0.79; 95% CI, 0.56–1.12; 1-sided *p* = 0.09); adjusted for crossover, the mOS with the placebo was 5.1 months (95% CI, 3.8–7.6; HR 0.49 [95% CI, 0.34–0.70]; 1-sided *p* < 0.001) [14]. Based on these results, in 2021, the FDA approved ivosidenib for adult patients with previously treated advanced IDH1-mutated cholangiocarcinoma [15].

More recently, in 2024, the FDA granted approval for vorasidenib, an oral brain-penetrant inhibitor of mutant IDH1 and IDH2 enzymes, for the treatment of adult and pediatric patients aged 12 and older with grade-2 astrocytoma or oligodendroglioma following surgery (including biopsy, subtotal resection, or gross total resection) and with a susceptible IDH1 or IDH2 mutation [16]. In the randomized phase III INDIGO trial, vorasidenib significantly improved PFS compared to the placebo (mPFS, 27.7 months vs. 11.1 months; HR 0.39; 95% CI, 0.27 to 0.56; *p* < 0.001); vorasidenib also delayed the time to next intervention (HR 0.26; 95% CI, 0.15 to 0.43; *p* < 0.001) [17] (Table 1).

Another promising target is represented by histone demethylases: in particular, lysine-specific demethylase 1 (LSD1/KDM1A), the first discovered histone lysine demethylase, can disrupt the molecular changes driving the epigenetic plasticity of cancer cells. LSD1 is implicated in various solid tumors, with its overexpression linked to poor prognosis. Recently, LSD1-inhibitors have entered clinical trials for cancer treatment, and results are highly awaited [18].

## 3. Combining Epigenetic Drugs with Other Anti-Cancer Therapies: Does More Mean Better?

Preclinical evidence has shown that combining epigenetic drugs with other anticancer therapies can enhance therapeutic efficacy and reduce drug resistance, leading to the investigation of many combinations with epigenetic drugs.

### 3.1. Combination with Chemotherapy

Chemoresistance, often linked to abnormal DNA methylation and alterations in histone acetylation and methylation, can potentially be reversed by epigenetic drugs. On the other hand, preclinical evidence suggests that HDAC inhibitors may enhance the effectiveness of chemotherapeutic agents by promoting chromatin decondensation, potentially leading to a synergistic therapeutic effect [19,20]. However, despite this promise, several clinical trials investigating the combination of epigenetic drugs and chemotherapy across various solid tumors have yielded disappointing results, with many trials being discontinued due to lack of efficacy. Indeed, although a phase II trial investigating carboplatin, paclitaxel, and vorinostat in advanced NSCLC patients showed a trend toward improved ORR with the addition of vorinostat (NCT 00481078) [21], the subsequent phase III trial in the same population failed to meet the primary endpoint of overall survival (OS) in a pre-specified interim analysis, revealing a negative impact of vorinostat on clinical outcomes (NCT00473889) [22]. Similarly, a phase II study of gemcitabine combined with the HDAC inhibitor tacedinaline (CI-994) in advanced pancreatic ductal adenocarcinoma (PDAC) showed no significant improvement in OS, RR, or PFS compared to gemcitabine alone (NCT00004861) [23]. Moreover, despite preclinical evidence suggesting that DNA methylation may contribute to the development of platinum resistance in ovarian cancer (OC), a randomized phase II trial evaluating the combination of the DNA-hypomethylating agent decitabine and carboplatin in recurrent, partially platinum-sensitive cases was terminated due to toxicity and a detrimental effect, irrespective of methylation status (NCT00748527) [24]. Additionally, a phase II trial evaluating the second-generation DNA-hypomethylating agent, guadecitabine, in combination with carboplatin in platinum-resistant ovarian cancer did not result in statistically significant improvements in clinical outcomes. (NCT01696032) [25] (Table 2).

### 3.2. Combination with Radiotherapy

The potential of epigenetic drugs as radiation sensitizers has been demonstrated in preclinical research by disrupting DDR and the cell cycle and increasing oxidative stress [26] and subsequently investigated in many clinical trials, usually concurrently with radiotherapy. Unfortunately, the combination of epigenetic drugs with radiotherapy (RT) in clinical trials has mostly resulted in increased levels of toxicity and minimal patient benefit. A significant challenge lies in identifying the most effective combination and timing for administering epigenetic drugs alongside radiotherapy. While some trials combining RT with epigenetic drugs showed no benefit [27], others have provided encouraging results. A phase I study combining the HDAC inhibitor vorinostat with capecitabine and radiotherapy in patients with localized pancreatic ductal adenocarcinoma (PDAC) was well tolerated, resulting in four R0 resections among 11 patients who underwent surgical exploration and a median overall survival (OS) of 1.1 years (NCT00983268) [28]. Additionally, a phase I trial demonstrated that the combination of vorinostat with vectorized internal radiotherapy using 131I-metaiodobenzylguanidine (MIBG) was tolerable in children with relapsed or refractory high-risk neuroblastoma, achieving an overall response rate (ORR) of 12% across all dose levels and 17% at the recommended phase II dose (NCT01019850) [29]. In the subsequent randomized phase II trial, patients with relapsed or refractory neuroblastoma receiving MIBG plus vorinostat had the highest response rate (32%) compared to 14% in the other treatment arms, meeting the prespecified threshold, with manageable toxicity; in contrast, vincristine and irinotecan did not enhance the response rate to MIBG and were associated with increased toxicity [30]. (Table 3). These findings highlight the need for implementing biomarkers to guide patient selection, since a key challenge lies in identifying the most effective combination and timing for administering epigenetic drugs alongside radiotherapy.

### 3.3. Combination with Hormone Therapy

The combination of epigenetic drugs with hormone therapy has been explored in various cancers, particularly breast and prostate cancer. In breast cancer, HDAC inhibitors have long shown the ability to interfere with estrogen-receptor-signaling pathways in estrogen-receptor-positive (ER+) breast cancer (BC) [31]. A phase II trial combining vorinostat with tamoxifen in hormone-therapy-resistant BC reported a 19% ORR and a 40% clinical benefit rate, demonstrating the potential to reverse hormone resistance [32]. The ENCORE 301 phase II trial, which combined entinostat with exemestane in postmenopausal women with advanced hormone-receptor-positive, endocrine-resistant BC showed an improvement in PFS [33]. However, the subsequent phase III E2112 trial failed to demonstrate a significant survival benefit in the same population, with a median OS of 23.4 months for the combination versus 21.7 months for exemestane alone (NCT02115282) [34]. Conversely, the phase III ACE study evaluating tucidinostat with exemestane in postmenopausal patients with advanced hormone-receptor-positive BC showed a significant benefit in PFS (NCT02482753) [35].

In prostate cancer, a phase I/II trial combining the HDAC inhibitor panobinostat with the anti-androgen bicalutamide in castration-resistant prostate cancer (CRPC) demonstrated improved outcomes if compared to historical controls, though high doses of panobinostat were associated with significant toxicity, mostly thrombocytopenia [36]. BET inhibitors, which also target androgen receptor signaling, have shown promise in preclinical CRPC models, including enzalutamide-resistant cases [37]. In a phase Ib/II study, the combination of the BET inhibitor ZEN-3694 with enzalutamide showed acceptable tolerability and potential efficacy in patients with androgen-signaling inhibitors-resistant metastatic CRPC [38]. However, a phase Ib study of the BET inhibitor GS-5829 combined with enzalutamide showed limited efficacy [39]. Another ongoing trial is investigating the novel BET inhibitor NUV-868, both as monotherapy and in combination with olaparib or enzalutamide, across various cancers including prostate cancer (NCT05252390) [40] (Table 4).

### 3.4. Combination with Targeted Therapy

The use of epigenetics drugs has been explored in combination with targeted therapies in different settings. Preliminary evidence has demonstrated that the use of epigenetic drugs in combination has the potential to overcome or delay the appearance of mechanisms of resistance, which are the most common reason for a target therapy failure. Here, we report significant results in combinations with epidermal growth factor receptor tyrosine kinase family (ErbB) inhibitors, anti-angiogenic agents, mammalian target of rapamycin (mTOR) inhibitors and PARP inhibitors.

The synergism between HDAC inhibitors and ErbB inhibitors has shown divergent results. The synergism between HDAC inhibitors and ErbB inhibitors was explored in a randomized phase II trial comparing first-generation EGFR inhibitor erlotinib with or without entinostat, an HDAC inhibitor, in 132 patients with advanced and treatment-naive NSCLC. Primary endpoint was 4-month PFS, which was not met (20% with erlotinib and 18% with erlotinib and entinostat, *p* = 0.7) (NCT00602030) [41]. Similar disappointing results comes from a randomized, placebo-controlled, double-blind phase IIb trial of first-generation EGFR inhibitors (gefitinib or erlotinib) with or without nicotinamide (an HDAC3 inhibitor) in 110 patients with an EGFR-mutant NSCLC; after a median follow-up of 54.3 months, the median PFS and OS were similar between the nicotinamide and the control group (mPFS: 12.7 m vs. 10.9 m, *p* = 0.2; mOS: 31.0 m vs. 29.4 m, *p* = 0.2); however, in a subgroup analysis, a significant reduction in mortality risk was observed in the nicotinamide group for females (*p* = 0.01) and never smokers (*p* = 0.03) [42].

The combination between HDAC inhibitors and anti-angiogenic agents was also evaluated. A phase I trial explored the combination of vorinostat and bevacizumab in patients affected by clear cell renal cell carcinoma (ccRCC): the combined treatment was tolerable, the ORR was 18%, with 5 partial (PR) and 1 complete response (CR), and the median PFS and OS were 5.7 months and 13.9 months, respectively [43]. Currently, in the same setting, a randomized, double-blind, placebo-controlled phase III trial (RENAVIV) is investigating the combination of pazopanib plus the potent oral pan-HDAC inhibitor abexinostat versus pazopanib alone in patients with advanced ccRCC and without prior exposure to anti-angiogenic TKIs (NCT03592472) [44]. In hepatocellular carcinoma (HCC), resminostat was tested alone or in combination with sorafenib in a phase I/II trial with patients progressing to first-line sorafenib: the combination was safe, and the PFS rate after 12 weeks of treatment (primary endpoint) was 12.5% in the resminostat group and 62.5% in the combination group [45]. Also, the HDAC inhibitor panobinostat was evaluated with bevacizumab in a phase I study with patients affected by high-grade glioma, with promising results (3 PR and 7 SD) and a good tolerability profile [46]. However, a subsequent phase II trial that compared panobinostat plus bevacizumab to bevacizumab alone in patients with recurrent glioblastoma or anaplastic glioma failed to demonstrate a significant benefit in terms of 6 months PFS for the combination arm [47].

The combination between HDAC inhibitors and mTOR inhibitors has been explored in a few early-phase studies. In a phase I trial, the association between the HDAC inhibitor vorinostat and the mTOR inhibitor ridaforolimus showed signals of efficacy in advanced solid tumors, especially ccRCC, despite thrombocytopenia as a significant toxicity [48]. Similarly, another phase I trial showed comparable results with the combination of vorinostat and the mTOR inhibitor sirolimus [49]. Conversely, a single-arm phase II trial exploring the combination of the HDAC inhibitor panobinostat and the mTOR inhibitor everolimus in children and young adults with advanced or recurrent gliomas harboring H3.1 or H3.3 K27M mutation has been withdrawn due to low accrual (NCT03632317) [50] (Table 5).

Due to the uncontrolled growth of cancer cells, DNA damage occurs, making the inhibition of DNA repair proteins a potential therapeutic strategy to effectively disrupt DNA repair mechanisms in cancer cells and thereby enhance cell death. A rationale for testing combination treatments with HDAC and PARP inhibition in cancers not sensitive to PARP inhibitor monotherapy has been defined, given that HDAC inhibition has been shown to be able to induce pharmacologic “BRCAness” in cancer cells with proficient DNA repair activity [51,52]. In BRCA-proficient high-grade serous ovarian and triple-negative breast cancer models, in vivo guadecitabine plus talazoparib treatment decreased xenograft tumor growth and increased overall survival, supporting clinical rationale for PARP inhibitors and epigenetic drugs combinations [53]. A Phase I/Ib clinical trial is currently underway, recruiting to evaluate the safety and preliminary efficacy of the PARP inhibitor olaparib and the HDAC inhibitor vorinostat combination in patients with relapsed, refractory, or metastatic breast cancer (NCT03742245) [54]. Similarly, a phase II trial is exploring the efficacy of ZEN003694, a BET inhibitor, plus talazoparib in patients with recurrent ovarian cancer (NCT05071937) [55].

### 3.5. Combination with Immunotherapy

Epigenetic drugs can potentially be exploited to modulate antitumor immunity [56,57], and there is increasing evidence that combining epigenetic with immunotherapeutic drugs may be beneficial to overcome acquired resistance to immunotherapy, for example, with DNMT and HDAC inhibitors [58]. Epigenetic drugs can act both on cancer cells and on cells involved in the immune response, exerting a potential role of modulators for immunotherapy [59]. Nevertheless, a randomized phase II study revealed an increased risk of toxicities and no therapeutic benefit from the combination of anti-PD1 pembrolizumab with the second-generation DNMT inhibitor CC-486 (oral azacytidine) [60]. Similarly, only a limited benefit was reported with CC-486 and durvalumab in immunologically cold tumors [61]. Recent results proved that guadecitabine in combination with pembrolizumab can be tolerable and harbors anticancer activity in a phase I dose-escalation study in patients with advanced solid tumors; overall, thirty patients were evaluable for antitumor activity: ORR was 7%, with 37% achieving disease control for 24 weeks or more; of 12 evaluable patients with non-small cell lung cancer (NSCLC), 10 had been previously treated with immune checkpoint inhibitors with 5 patients (42%) who reported a disease control equal or longer than 24 weeks [62]. This may favor the hypothesis of a reversibility of resistance to immunotherapy with the integration of epigenetic drugs [62]. Another study tested the combination of HDAC inhibitor romidepsin with cisplatin and nivolumab in a phase I/II study (NCT02393794) in metastatic TNBC; this association showed a good safety profile: the ORR was 44%, median PFS was 4.4 months, and 1-year PFS rate was 23%; the median OS was 10.3 months [63]. In metastatic melanoma, guadecitabine combined with anti-CTLA4 ipilimumab resulted in an immune-related DCR of 42% and ORR of 26% in the phase Ib NIBIT-M4 study (NCT02608437) [64]. Also, in metastatic melanoma, the addition of entinostat to pembrolizumab (ENCORE-601) demonstrated an ability to restore inflammation in the tumor microenvironment (TME), representing a positive assumption for successful re-treatment with anti-PD-1/PD-L1 [65]. In a phase II trial (NCT02538510), the combination of pembrolizumab and vorinostat in recurrent or metastatic head and neck squamous cell carcinoma (HNSCC) and salivary gland cancer (SGC): partial responses were reported both in HNSCC and in SGC. Among 25 patients with HNSCC, 8 (32%) achieved PR, and 5 (20%) had SD; the median OS was 12.6 months, and the median PFS was 4.5 months; among SGC patients, 4 (16%) out of 25 patients had PR, and 14 (56%) had SD; the median OS was 14 months, and the median PFS was 6.9 months [66]. A pilot study of tazemetostat and pembrolizumab in advanced urothelial carcinoma (ETCTN 10183) showed tolerability; moreover, activity was seen with PR in 3 pts (25%) and SD in 3 (25%). The median PFS was 3.1 months (95%CI: 2.3–NA), and the median overall survival was 8.0 months (95% CI: 4.7–NA) [67]. Hopefully, results from the ongoing study will help define the depth of the degree of reversibility of resistance to immunotherapy. A phase II study of nivolumab plus ipilimumab and ASTX727 versus nivolumab plus ipilimumab in anti-PD-1/PD-L1-resistant melanoma or NSCLC patients is ongoing [68] (Table 6). Finally, future and ongoing trials may elucidate the possible long-term effects of the prolonged use of epigenetic drugs in combination with immunotherapy, including the potential T-cell depletion, which may be induced, for example, by BET inhibitors [69].

## 4. Current Challenges and Future Directions

The emergence of new precision approaches may represent a transformational force to unlock the potential of the epigenetic therapeutics field, which has been marked by modest success and notable disappointments so far. Historically, one of the main problems of epigenetic drugs has been represented by high toxicities and low specificity [70]. As emphasized by the recent FDA-granted accelerated approval of vorasidenib in low-grade glioma with susceptible IDH1/2 mutations [16], as well as ivosidenib for adult patients with previously treated advanced cholangiocarcinoma with IDH1 mutations [15], epigenetic drugs are becoming increasingly specific for their target enzymes and, thus, their development should follow a precision-medicine approach. Exploration of lower doses and targeted delivery might improve the therapeutic index of epigenetic drugs [71].

To maximize the potentiality of epigenetic drugs, trial design structure, endpoints selection, and dose administration should be carefully determined. Moreover, according to the mechanism of action of the epigenetic treatment, dosage, and schedule of epigenetic drugs should be adequately selected; for example, intermittent dosing might represent a means to reduce severe toxicities [72]. Moreover, long-term follow-up with recording of late responses or clinical benefit in terms of prolonged objective responses or stable disease may be considered to evaluate tumor growth rate [71].

Indeed, to accurately assess the true effectiveness of epigenetic agents, molecular biomarkers, tumor markers, and extended follow-up of PFS and OS may represent alternative outcome measures or epigenetic drug activities, due to the extended time required to modulate epigenetic activity [73], and given that, any clinical trial design should be carefully justified according to the characteristics of drug and population of interest [74].

The traditional dose escalation method has shown itself to be imperfect in earlier trials for determining the optimal dosage and scheduling of epigenetic agents, especially when the goal is to leverage their epigenetic-modifying potential rather than their cytotoxic effects [73]. Combinations with epigenetic drugs have been mostly impaired by the occurrence of acute or chronic dose-limiting toxicities (DLTs); according to a single-center retrospective study, 66% of grade-3 or grade-4 epigenetic drug-related toxicities occurred after the first cycle, advocating for the need to prolong the assessment period [75]. Presumably, cytotoxic effects may not represent the ultimate of these drugs; therefore, the time required for epigenetic drugs to reprogram transcriptional activities and trigger cell differentiation or phenotypic changes in clinical tissues remains a matter of preclinical and clinical investigation [71]. Evaluating biomarker response, methylation, and expression differences between normal and tumor tissues in translational studies should assist the quantitative assessments to determine the therapeutic window of epigenetic drugs [76].

## Figures and Tables

**Figure 1 ijms-25-11740-f001:**
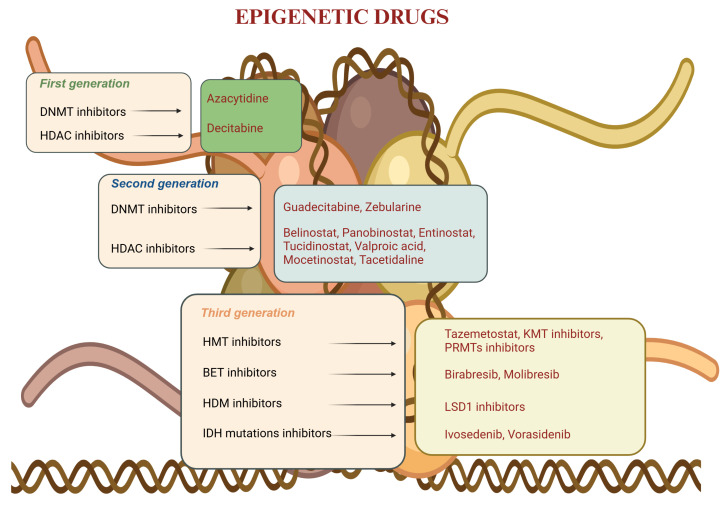
Generations of epigenetic drugs and key compounds [5]. BET: bromodomain and extra-terminal domain; DNMT: DNA methyltransferase; HDAC: histone deacetylase; HDM: histone demethylase; HMT: Histone methyltransferase; KMT; LSD: lysine-specific histone demethylase; PRMT: protein arginine methyltransferase inhibitor.

**Table 1 ijms-25-11740-t001:** Key clinical trials for FDA-approved epigenetic drugs as monotherapy in solid tumors.

Clinical Trial	Phase	Number of Patients	Type of Solid Tumors	Drug	Results
**NCT02601950** [11]	II	62	advanced epithelioid sarcoma with loss of INI1/SMARCB1	tazemetostat (EZH2 inh)	**ORR**: 15% (95% CI 7–26); **Median PFS**: 5.5 mo (95% CI 3.4–5.9); **Median OS**: 19.0 mo (11.0–NE).
**NCT02989857** [14]	III	185	IDH1-mutated advanced cholangiocarcinoma	ivosidenib (IDH1 inh)	**Median PFS**: 6.9 mo (ivosidenib) vs. 1.4 mo (placebo), HR 0.37, 95% CI, 0.25–0.54, *p* < 0.001. **Median OS**: 10.3 mo (ivosidenib) vs. 7.5 mo (placebo), HR 0.79, [95% CI, 0.56–1.12]; 1-sided *p* = 0.09.
**NCT04164901** [17]	III	331	IDH1 or IDH2 mutated low-grade glioma	vorasidenib (IDH1/IDH2 inh) vs. placebo	**Median PFS**: 27.7 mo vs. 11.1 mo (HR 0.39; 95% CI, 0.27 to 0.56; *p* < 0.001)

CI: confidence interval; EZH2: enhancer of zeste homologue 2; IDH: isocitrate dehydrogenase; inh: inhibitor; HR: hazard ratio; mo: months; NE: not estimable; ORR: objective response rate; OS: overall survival; PFS: progression-free survival.

**Table 2 ijms-25-11740-t002:** Key clinical trials of epigenetic drugs combined with chemotherapy.

Trial	Phase	Number of Patients	Type of Solid Tumors	Drugs	Results
**NCT00481078** [21]	II	94	advanced NSCLC	CBDCA + TXL + vorinostat vs. CBDCA + TXL + placebo	**ORR** (primary endpoint): 34% (CBDCA + TXL + vorinostat) vs. 12%. (CBDCA + TXL + placebo), *p* = 0.02
**NCT00473889** [22]	III	253	advanced NSCLC	CBDCA + TXL + vorinostat vs. CBDCA + TXL + placebo	**OS** (primary endpoint): 11 mo (0.2 to 17.3) (arm vorinostat + CBDCA + TXL) vs. 14 mo (0.03 to 18.7) (arm placebo+CBDCA + TXL), *p* = 0.992
**NCT00004861** [23]	II	174	advanced PDAC	gemcitabine + tacedinaline (CI-994) vs. gemcitabine + placebo	**Median OS** (primary endpoint): 194 days (gemcitabine + tacedinaline) vs. 214 days (gemcitabine + placebo), *p* = 0.908
**NCT00748527** [24]	II	29	advanced OC with or without methylated hMLH1	CBDCA (arm A) vs. decitabine + CBDCA (arm B)	**ORR** by GCIG criteria in methylated hMLH1 tumor (primary endpoint): Responses (PR/CR) in 9/14 patients (arm A) vs. 3/15 patients (arm B). **ORR** regardless of methylation status (secondary endpoint): Responses (PR/CR) in 7/13 patients (arm A) vs. 1/12 patients (arm B).
**NCT01696032** [25]	II	100	Platinum-resistance-advanced OC	guadecitabine + carboplatin vs. treatment of choice (topotecan, pegylated liposomal doxorubicin, paclitaxel, or gemcitabine).	**Median PFS** (primary endpoint): 16.3 w (guadecitabine +carboplatin) vs. 9.1 w (treatment of choice); *p* = 0.07

CBDCA: carboplatin; NSCLC: non-small cell lung cancer; OC: ovarian cancer; ORR: objective response rate; OS: overall survival; PDAC: pancreatic ductal adenocarcinoma; PFS: progression-free survival; TXL: paclitaxel; w: weeks.

**Table 3 ijms-25-11740-t003:** Key clinical trials involving epigenetic drugs with radiotherapy.

Trial	Phase	Number of Patients	Type of Solid Tumors	Drugs	Results
**NCT00983268** [28]	I	21	non-metastatic PDAC	vorinostat + CAPE + RT	**MTD**: vorinostat 400 mg daily + CAPE 1000 mg BID (during RT). **ORR**: 90% SD, 10% PD (at time of surgery). **Resection**: R0 (4/11 patients) **Median OS**: 1.1y (95% CI 0.78–1.35).
**NCT01019850** [29]	I	27	relapsed and/or refractory HR neuroblastoma	vorinostat + vectorized internal radiotherapy with 131-I-MIBG.	**Safety**: Feasible and tolerable. **ORR**: 12% at all dose levels and 17% at the RP2D
**NCT02035137** [30]	II	114	relapsed or refractory neuroblastoma	Arm A: MIBG. Arm B: MIBC + vincristine + irinotecan. Arm C: vorinostat + MIBG	**ORR** (after 1 course- primary endpoint): 32% (MIBG + vorinostat) vs. 14% (other arms).

BID: bis in die; CAPE: capecitabine; CI: confidence interval; HR: high-risk; 131-I-MIBG: metaiodobenzylguanidine; MTD: maximum tolerated dose; ORR: objective response rate; OS: overall survival; PD: progressive disease; PDAC: pancreatic ductal adenocarcinoma; RP2D: recommended phase II dose; RT: radiotherapy; SD: stable disease; Y: year.

**Table 4 ijms-25-11740-t004:** Key clinical trials involving epigenetic drugs with hormone therapy.

Trial	Phase	Number of Patients	Type of Solid Tumors	Drugs	Results
**NCT00365599** [32]	II	43	ER-positive, hormone therapy-resistant mBC	vorinostat + tamoxifen	**ORR**: 19%. **CBR** (response or stable disease > 24 w) 40%; **Median DOR**: 10.3 mo (CI: 8.1–12.4).
**NCT00676663** [33]	II	130	postmenopausal advanced ER +BC	EXE + entinostat vs. EXE + placebo	**Median PFS**: 4.28 mo (EXE + entinostat) vs. 2.27 mo (EXE + placebo), HR 0.73, *p* = 0.06.
**NCT02115282** [34]	III	608	advanced HR + BC	EXE + entinostat vs. EXE + placebo	**Median PFS**: 3.3 mo (EXE + entinostat) vs. 3.1 mo (EXE + placebo), HR 0.87; *p* = 0.30; **Median OS**: 23.4 mo (EXE + entinostat) vs. 21.7 mo (EXE + placebo), HR 0.99, *p* = 0.94
**NCT02482753** [35]	III	365	advanced HR + BC	EXE + tucidinostat vs. EXE + placebo	**Median PFS**: 7.4 mo (tucidinostat) vs. 3.8 mo (placebo), HR 0.75, *p* = 0.033.
**NCT00878436** [36]	I/II	55	mCRPC	panobinostat + bicalutamide Arm A: 40 mg panobinostat Arm B: 20 mg panobinostat	**36W-PFS**: 47.5% (arm A) vs. 38.5% (arm B).
**NCT02711956** [38]	Ib/IIa	75	mCRPC	ZEN-3694 + ENZA	**PFS**: 9 mo (95% CI 4.6–12.9)

CBR: clinical benefit rate; CI: confidence interval; DOR: duration of response; ENZA: enzalutamide; ER: estrogen receptor; EXE: exemestane; HR+: hormone-receptor-positive; mBC: metastatic breast cancer; mCRPC: metastatic castration-resistant prostate cancer; mo: months; ORR: objective response rate; OS: overall survival; PFS: progression-free survival; w: weeks.

**Table 5 ijms-25-11740-t005:** Key clinical trials involving epigenetic drugs combined with targeted therapy.

Trial	Phase	Number of Patients	Type of Solid Tumors	Drugs	Results
**NCT02416739** [42]	IIb	110	EGFR-mutated NSCLC	erlotinib/gefitinib + nicotinamide vs. erlotinib/gefitinib + placebo	**mPFS** (primary endpoint): no statistically significant difference (12.7 m vs. 10.9 m, *p* = 0.2). Subgroup analysis: significant reduction in mortality risk in females and non-smokers.
**NCT00324870** [43]	I/II	36	ccRCC	BEV + vorinostat	**Feasible and tolerable.** **ORR**: 18% **mPFS**: 5.7 mo **mOS**: 13.9 mo
**NCT00943449** [45]	I/II	57	HCC	resminostat +/− sorafenib	**12w-PFS** (primary endpoint): 62.5% with the combination, 12.5% with sorafenib. **Feasible and tolerable.**
**NCT00859222** [46]	I	10	high-grade glioma	BEV + panobinostat	**ORR**: 30% PR; 70% SD.

BEV: bevacizumab; ccRCC: clear cell renal cell carcinoma; HCC: hepatocellular carcinoma; NSCLC: non-small cell lung cancer; ORR: objective response rate; OS: overall survival; PFS: progression-free survival; PR: partial response; SD: stable disease.

**Table 6 ijms-25-11740-t006:** Key clinical trials involving epigenetic drugs combined with immunotherapy.

Trial	Phase	Number of Patients	Type of Solid Tumors	Drugs	Results
**NCT02546986** [60]	II	100	NSCLC	azacitidine (CC-486) + PEMBRO	**Median PFS** (primary endpoint) 2.9 mo (PEMBRO + CC-486) vs. 4.0 mo (PEMBRO + placebo)
**NCT02393794** [63]	I/II	51	locally recurrent or metastatic TNBC	romidepsin + CDDP + NIVO	**ORR**: 44%; **Median PFS**: 4.4 mo; **1-year-PFS**: 23%; **Median OS**: 10.3 mo.
**NCT02538510** [66]	I/II	25 HNSCC 25 SGC	recurrent/metastatic HNSCC or salivary gland cancer	vorinostat + PEMBRO	Primary endpoints were **safety** and **ORR**: - in HNSCC: CR = 0, PR = 8 (32%), SD = 5 (20%). - in SGCs: CR = 0, PR = 4 (16%), SD = 14 (56%)
**NCT03854474** [67]	I/II	12	advanced UC	tazemetostat + PEMBRO	Primary endpoint: **safety**: no DLTs. **ORR**: PR in 3 patients (25%), SD in 3 patients (25%). **Median PFS**: 3.1 months (95%CI: 2.3–NA); **Median OS**: 8.0 months (95% CI: 4.7–NA).

CDDP: cisplatin; CR: complete response; DLTs: dose-limiting toxicities; HNSCC: head and neck squamous cell carcinoma; mo: months; OS: overall survival; PFS: progression-free survival; NA: not available; NIVO: nivolumab; NSCLC: non-small cell lung cancer; ORR: objective response rate; OS: overall survival; PEMBRO: pembrolizumab; PFS: progression-free survival; PR: partial response; SD: stable disease; SGC: salivary gland cancer; TNBC: triple-negative breast cancer; UC: urothelial carcinoma; w: weeks.

## Data Availability

The data used to support the findings of this study are included within the article.

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
