# Peer review of "Progresses and Pitfalls of Epigenetics in Solid Tumors Clinical Trials"

_ijms, 2024, doi:10.3390/ijms252111740_

Round 1

Reviewer 1 Report

Comments and Suggestions for Authors

In the context of Table 1 - The difference between first and second generation epigenetic drugs need to be made very clear for the benefit of readers with not enough background knowledge.

For the third generation epigenetic drugs, is it possible to re-align them as inhibitors of methylation or acetylation, so that a connection with first/second generation drugs is apparent ? – perhaps adding a column to mention this activity will be helpful.

Also, in context of Table 1, it is not apparent whether these drugs have only been tried in vitro or in clinical trials and also which cancers – all this information is needed to make the Table more meaningful.

In Table 2, authors only list 3 clinical trials and at the Introduction of this table, they mention - ‘This class of drugs resulted in improved efficacy, leading in few cases to the Food and Drug Administration (FDA) approval in solid tumors’. So, my query is whether these 3 studies are the only instances of FDA approval? If so, please clearly mention. Also, it will be helpful if all the clinical trials are listed so that readers have an idea that out of x clinical trials, only 3 lead to FDA approval. This will provide a more comprehensive knowledge on the topic. When doing so, please do not forget to highlight these 3 trials to mention that these 3 led to FDA approval.

Does the Table 3 provide a comprehensive compilation of all trials that test combination of epigenetic drugs with chemotherapy – total 4 trials? Please check and provide all information here.

This reviewer notices that the information discussed in the text regarding different clinical trials is not very detailed and is in many cases just a little more elaboration of what is already provided in the respective Tables. This makes it a very boring to read with not much background, rationale etc.

Section 3.6 on combination with PARP inhibitors is not very exhaustive and can be combined with one of the earlier sections.

All the information provided in this review article is in the form of many Tables. Authors should add an informative cartoon diagram that discusses the principle and the major developments (include mechanistic details) in the field of clinical trials of epigenetic drugs.

Author Response

Dear reviewer,

First of all, we would like to thank you for your precious comments. Here below our answers that you will find in the new updated version of the review.

COMMENT 1. In the context of Table 1 - The difference between first and second generation epigenetic drugs need to be made very clear for the benefit of readers with not enough background knowledge. 

COMMENT 2. For the third generation epigenetic drugs, is it possible to re-align them as inhibitors of methylation or acetylation, so that a connection with first/second generation drugs is apparent ? – perhaps adding a column to mention this activity will be helpful.

COMMENT 3. Also, in context of Table 1, it is not apparent whether these drugs have only been tried in vitro or in clinical trials and also which cancers – all this information is needed to make the Table more meaningful.

-> RESPONSE TO COMMENT 1, 2, 3:  Table 1 was changed. A figure (Figure 1) was added, which should also be more intuitive for the reader. Moreover, in the text we better differentiated generations of epigenetic drugs.

COMMENT 4. In Table 2, authors only list 3 clinical trials and at the Introduction of this table, they mention - ‘This class of drugs resulted in improved efficacy, leading in few cases to the Food and Drug Administration (FDA) approval in solid tumors’. So, my query is whether these 3 studies are the only instances of FDA approval? If so, please clearly mention. Also, it will be helpful if all the clinical trials are listed so that readers have an idea that out of x clinical trials, only 3 lead to FDA approval. This will provide a more comprehensive knowledge on the topic. When doing so, please do not forget to highlight these 3 trials to mention that these 3 led to FDA approval.

-> RESPONSE TO COMMENT 4: A table (Table 1) with references deals with FDA-approved drugs in solid tumors. In the free text, this has been better highlighted.

COMMENT 5. Does the Table 3 provide a comprehensive compilation of all trials that test combination of epigenetic drugs with chemotherapy – total 4 trials? Please check and provide all information here.

-> RESPONSE TO COMMENT 5: We re-checked the literature for latest advancements in the field; we also changed the structure of the tables to allow the reader to identify key clinical trials regarding the topic of the paragraph.

COMMENT 6. This reviewer notices that the information discussed in the text regarding different clinical trials is not very detailed and is in many cases just a little more elaboration of what is already provided in the respective Tables. This makes it a very boring to read with not much background, rationale etc.

-> RESPONSE TO COMMENT 6: tables have been differentiated from the content of the text, making it more interesting to read the text, on one hand. On the other hand, the reader may appreciate the tables to identify the key trials for each category of combination therapy.

COMMENT 7. Section 3.6 on combination with PARP inhibitors is not very exhaustive and can be combined with one of the earlier sections.

-> RESPONSE TO COMMENT 7: PARP inhibitors' section has been combined with targeted therapy.

COMMENT 8. All the information provided in this review article is in the form of many Tables. Authors should add an informative cartoon diagram that discusses the principle and the major developments (include mechanistic details) in the field of clinical trials of epigenetic drugs.

-> RESPONSE TO COMMENT 8: A figure was added. Moreover, in the text, along with different paragraphs, a more detailed explanation of mechanisms and rationale of combinations has been provided.

Reviewer 2 Report

Comments and Suggestions for Authors

This manuscript offers a thorough overview of the current landscape of epigenetic therapies in solid tumors, highlighting both advancements and challenges in the field. The authors effectively summarize key developments in drug design, trial methodologies, and the integration of precision medicine approaches. However, several areas require clarification, critical analysis, and expansion to enhance the manuscript's depth and impact. Below are some specific suggestions and queries:

>  The table contents included in the manuscript are not referenced appropriately. As this is a review paper, please ensure that all relevant table contents are cited with appropriate references.

>  The manuscript currently lacks figures or schematic diagrams, making it appear somewhat ordinary. Including visual elements would greatly enhance the reader’s understanding. Please incorporate relevant figures, ensuring to obtain proper permissions from publishers where necessary.

>  It is crucial for review papers to encompass the most relevant studies published in the last 3-5 years. This approach would provide readers with insights into the latest advancements in the field. Please consider expanding the literature review to include these recent works.

  By addressing these points, the authors can significantly improve the clarity and impact of their manuscript. I look forward to the revised submission. 

Author Response

Dear reviewer,

We would like to thank you for your comments. Here below our answers that you will find in the new updated version of the review.

COMMENT 1>  The table contents included in the manuscript are not referenced appropriately. As this is a review paper, please ensure that all relevant table contents are cited with appropriate references.

Thank you. Every cited key trial has been associated to a proper reference.

COMMENT 2>  The manuscript currently lacks figures or schematic diagrams, making it appear somewhat ordinary. Including visual elements would greatly enhance the reader’s understanding. Please incorporate relevant figures, ensuring to obtain proper permissions from publishers where necessary.

A figure (Figure 1) was added, which should also be more intuitive for the reader (proper permission was obtained)

COMMENT 3>  It is crucial for review papers to encompass the most relevant studies published in the last 3-5 years. This approach would provide readers with insights into the latest advancements in the field. Please consider expanding the literature review to include these recent works.

We re-checked the literature for latest advancements in the field. We also changed the structure of the tables to allow the reader to identify key clinical trials regarding the topic of the paragraph. Moreover, a more detailed explanation of mechanisms and rationale of combinations has been provided.

Round 2

Reviewer 1 Report

Comments and Suggestions for Authors

My concerns have been adequately addressed, and the manuscript can be accepted for publication.

Reviewer 2 Report

Comments and Suggestions for Authors

The authors have adequately addressed all raised concerns and comments.

Thanks!